# In Vitro Evaluation of Antioxidant and Cytokine-Modulating Activity of Tormentil Rhizome Extract and Its Microbial Metabolites in Human Immune Cells

**DOI:** 10.3390/ijms262211164

**Published:** 2025-11-19

**Authors:** Aleksandra Kruk

**Affiliations:** 1Department of Pharmaceutical Biology, Faculty of Pharmacy, Medical University of Warsaw, Banacha 1 Street, 02-097 Warsaw, Poland; akruk@wum.edu.pl; 2Microbiota Lab, Department of Pharmaceutical Microbiology and Bioanalysis, Faculty of Pharmacy, Medical University of Warsaw, Banacha 1b Street, 02-097 Warsaw, Poland

**Keywords:** *Tormentillae rhizoma*, *Potentilla erecta*, gut microbiota, postbiotic metabolites, neutrophil, macrophage, cytokine modulation, oxidative stress

## Abstract

The tormentil rhizome (*Potentilla erecta* L.) is traditionally used to treat gastrointestinal and inflammatory disorders, yet the mechanisms underlying its immunomodulatory activity remain unclear. No studies have examined the metabolism of tormentil constituents by the human gut microbiota and their effects on innate immune cells. This study evaluated the effects of the ethanolic extract of tormentil rhizome (EtTR) and its gut microbiota-derived metabolites (TRGMs) on innate immune function using human neutrophils and THP-1-derived macrophages. The chemical composition of EtTR and TRGMs was characterized by LC-MS, revealing fractions enriched in catechins and procyanidins (30% MeOH) or ellagic acid derivatives and triterpenes (100% MeOH). EtTR and all TRGM fractions significantly reduced ROS production, while the extract and selected metabolites decreased IL-1*β* and TNF-*α* secretion in neutrophils, whereas IL-8 showed marked induction. In macrophages, EtTR and selected fractions suppressed TNF-*α* and MCP-1 release but variably affected IL-6, reflecting donor-dependent modulation. The strongest inhibition was observed for fractions rich in catechins and triterpenoid conjugates, indicating synergistic activity between these compound classes. Overall, EtTR and its microbiota-derived metabolites exerted complementary antioxidant and immunomodulatory effects, providing mechanistic evidence that microbial transformation of tormentil polyphenols yields bioactive postbiotic metabolites capable of modulating inflammatory signaling.

## 1. Introduction

The gut microbiota is a crucial regulator of host immunity, redox balance, and inflammatory signaling. Polyphenols from plant-derived foods and herbal preparations are known to interact closely with intestinal microorganisms. Because of their high molecular weight and low intestinal absorption, these compounds undergo extensive microbial biotransformation in the colon, yielding smaller, more bioavailable metabolites that can exert systemic biological effects. This metabolic interplay, often referred to as the polyphenol–microbiota–immune axis, is increasingly recognized as a key determinant of the biological activity of tannin-rich herbal medicines [1,2,3].

*Tormentillae rhizoma* (the rhizome of *Potentilla erecta* L.) has a long history of use in European folk medicine, especially for the treatment of diarrhea, mucosal irritation, and inflammatory conditions of the gastrointestinal tract [4,5]. In Germany, tormentil rhizome is traditionally consumed as a digestive tincture, while in Ukraine, a honey-based tincture is popular as a household remedy for intestinal discomfort and inflammation [6]. Despite this long-standing traditional use, the molecular basis of tormentil’s biological activity and the contribution of its gut microbiota-derived metabolites remain insufficiently characterized [7].

Phytochemically, tormentil rhizome is exceptionally rich in polyphenolic compounds, particularly condensed and hydrolysable tannins (up to 20–25% of dry mass), including catechin oligomers, procyanidins, agrimoniin, and ellagic acid derivatives [8,9]. In addition to tannins, the rhizome contains notable amounts of pentacyclic triterpenes, such as tormentil triterpenes (tormentic acid, myrianthic acid, and cecropiacic acid) and their esters [6]. These lipophilic constituents have been shown to exhibit anti-inflammatory, antioxidant, and wound-healing properties, partly by inhibition of cyclooxygenase (COX) and lipoxygenase (LOX) enzymes, modulation of nuclear factor kappa-light-chain-enhancer of activated B cells (NF-κB) and mitogen-activated protein kinase (MAPK) signaling pathways, as well as upregulation of nuclear factor erythroid 2-related factor 2 (Nrf2)-mediated antioxidant defense mechanisms. Together, these polyphenolic and triterpenoid constituents form a complex phytochemical matrix responsible for the broad pharmacological potential of tormentil rhizome [5,10,11].

Recent studies have demonstrated that tormentil rhizome exhibits notable anti-inflammatory activity both in vitro and in vivo. Agrimoniin-rich fractions obtained from a 50% ethanol extract and enriched in agrimoniin, the main ellagitannin of *P. erecta*, inhibited COX-2 expression and reduced prostaglandin E_2_ (PGE_2_) production in UVB- or interleukin-1*α* (IL-1*α*)-stimulated HaCaT keratinocytes. In a subsequent clinical study, topical application of the same fraction significantly decreased UVB-induced erythema in healthy volunteers, confirming its anti-inflammatory potential in acute skin inflammation models [12].

Although these effects were not evaluated directly in immune cells, they provide strong evidence of downstream modulation of inflammatory mediators. Consistent anti-inflammatory activity has also been reported for other *Potentilla* species. Ethanol extracts of *P. supina* and *P. paradoxa* suppressed nitric oxide (NO), PGE_2_, tumor necrosis factor alpha (TNF-*α*), and interleukin-6 (IL-6) production in lipopolysaccharide (LPS)-stimulated RAW 264.7 macrophages by downregulating inducible nitric oxide synthase (iNOS) and COX-2 expression and inhibiting NF-κB and activator protein 1 (AP-1) signaling. Both extracts also reduced inflammation in vivo in mouse models of endotoxemia and gastritis [13,14]. Likewise, *P*. discolor ethanol extract suppressed NO, TNF-*α*, and monocyte chemoattractant protein-1 (MCP-1) secretion and decreased iNOS mRNA expression in macrophages [15]. These findings indicate that ethanol extracts across the *Potentilla* genus consistently attenuate macrophage-mediated inflammatory responses through the NF-κB/AP-1 → iNOS/COX-2 → NO/PGE_2_/TNF-*α*/IL-6 axis, providing a strong mechanistic rationale for *P. erecta*. Moreover, metabolites derived from ellagitannins, particularly urolithins formed by gut microbiota, have demonstrated anti-inflammatory effects in human and murine immune cells, including THP-1 (human monocytic leukemia cell line)- and RAW 264.7-derived macrophages, by attenuating TNF-*α* production and enhancing interleukin-10 (IL-10) secretion through activation of extracellular signal-regulated kinase 1/2 (ERK1/2) signaling. Notably, their glucuronide conjugates were inactive in these models [16]. Similarly, urolithins, the gut microbiota-derived metabolites of ellagitannins, have been shown to exert strong anti-inflammatory and antioxidant effects. Urolithin A suppresses NF-κB and MAPK signaling, reducing the expression of COX-2 and iNOS and the release of pro-inflammatory mediators such as TNF-*α*, IL-6, and PGE_2_ while enhancing antioxidant enzyme activity. These findings suggest that microbial conversion of ellagitannins may significantly contribute to the anti-inflammatory potential of *P. erecta* extracts [12,17,18,19,20].

Neutrophils and macrophages are pivotal components of innate immunity and play essential roles in the initiation, amplification, and resolution of inflammation. Activated neutrophils rapidly generate reactive oxygen species (ROS) by nicotinamide adenine dinucleotide phosphate (NADPH) oxidase, release proteolytic enzymes, form neutrophil extracellular traps (NETs), and secrete cytokines such as interleukin-8 (IL-8) and TNF-*α* [21,22,23]. Macrophages, including THP-1-derived macrophages, coordinate inflammatory signaling by phagocytosis and secretion of cytokines (IL-1*β*, IL-6, TNF-*α*) and regulate redox homeostasis and apoptosis through multiple intracellular signaling networks. Dysregulation of these processes leads to persistent inflammation and tissue damage, contributing to the pathogenesis of chronic inflammatory and autoimmune diseases [24,25,26].

Although several studies have reported antioxidant, anti-inflammatory, and cytoprotective properties of tormentil rhizome extracts [4,5], the contribution of gut microbiota-derived metabolites and triterpenoid components to the modulation of immune cell function remains poorly defined. Moreover, no studies have yet compared the effects of the native ethanolic extract and its gut microbiota-derived metabolites on human innate immune cells. Published studies mainly concern other *Potentilla* species or focus exclusively on ellagitannin fractions, without considering metabolite fractions containing other compound classes. To address this gap, the study focused on two complementary models of innate immune response. Human neutrophils were selected as a model of acute inflammation due to their rapid response to inflammatory stimuli, involving ROS generation, cytokine release, and regulated cell death [23]. THP-1-derived macrophages, in contrast, represent a model of chronic or sustained inflammation and cytokine-driven activation, allowing for evaluation of metabolic activity and mediator secretion [27]. Together, these two models provide a comprehensive view of how tormentil rhizome and its gut microbiota-derived metabolites influence key processes in innate immune regulation.

The aim of this study was to evaluate and compare the immunomodulatory effects of the ethanolic extract of tormentil rhizome (EtTR) and its gut microbiota-derived metabolites (TRGMs) on human innate immune cells, specifically neutrophils and THP-1-derived macrophages, to elucidate their potential roles in modulating inflammation.

## 2. Results

### 2.1. Chemical Composition of Extract

Based on the comprehensive phytochemical study of the tormentil rhizome extract previously published by the author [6,28], which provided detailed structural characterization and quantitative data of its constituents, the present work includes only a summarized qualitative and quantitative profile to support the interpretation of the biological results obtained for EtTR and its gut microbiota-derived metabolites (Table 1). The ethanolic extract of *Tormentillae rhizoma* was selected for the present studies, as ethanol-based tinctures represent the most traditional form of tormentil rhizome use and ensure efficient extraction of both polyphenolic and triterpenoid constituents [6,29].

The raw extract was dominated by flavan-3-ols and their oligomeric products, which together accounted for approximately 68% of all quantified compounds. This group included catechin, its *O*-hexoside, and a range of oligomeric derivatives such as B-type procyanidin dimers, trimers, tetramers, and pentamers, indicating a high degree of polymerization typical of tormentil rhizome tannins. Triterpenoids and their glycosides constituted the second most abundant class (≈29%), represented mainly by tormentic acid, its *O*-hexoside isomers, and related ursane-type derivatives such as myrianthic and cecropiacic acids. Minor components included ellagic acid and its glycosides (≈2%), as well as chalcones (e.g., phlorizin, ≈1%) and phenolic acids (e.g., protocatechuic acid derivative). In addition, several unknown compounds with distinct *m*/*z* values were detected, reflecting the chemical diversity of the extract.

This compositional overview illustrates that EtTR is particularly rich in hydrolyzable and condensed tannins, as well as triterpenoid acids, both classes being known for their anti-inflammatory, antioxidant, and barrier-protective activities. Therefore, the chemical composition summarized here provides an essential background for understanding the biological effects of the extract and its gut-derived metabolites described in the following sections.

### 2.2. Chemical Composition of Gut Metabolites

The postbiotic mixtures were obtained after 24 h anaerobic incubation of the ethanolic extract of *Tormentillae rhizoma* with fecal slurries derived from three healthy human donors. Following incubation, the samples were subjected to solid-phase extraction (SPE) and eluted sequentially with 30% and 100% methanol, yielding two metabolite-containing fractions (TRGM_30 and TRGM_100) for each donor (D1–D3). A detailed chemical characterization of these metabolite fractions has been previously published by the author [28], and the present data are shown here solely to facilitate the interpretation of the subsequent biological findings (Figure 1 and Figure 2).

In the obtained metabolite fractions, several native compounds from the extract remained untransformed and were detected in the resulting metabolite fractions (Figure 1). Catechins and procyanidins (e.g., catechin, procyanidin dimer type B isomer I, procyanidin trimer type C isomer II) were predominantly detected in the 30% MeOH fractions, while the 100% MeOH fractions were enriched in ellagic acid derivatives (ellagic acid, methylellagic acid *O*-pentoside isomer II) and glycosylated triterpenoids such as tormentic acid *O*-hexoside and its isomers, dihydroxy-3-oxo-urs-12-en-28-oic acid hexosides, and cecropiacic acid. In addition, several unidentified metabolites (*m*/*z* 493 and 507) were consistently observed across all donors. Despite a similar qualitative profile, quantitative differences were observed between donors (D1–D3). Donor 3 exhibited the highest overall signal intensities for both catechin-type and triterpenoid derivatives, whereas donors 1 and 2 showed lower yet comparable levels. In the 30% MeOH fraction, D1 and D3 exhibited significantly higher catechin levels compared with D2, whereas procyanidin trimer type C isomer II was most abundant in D3, with D1 and D2 showing comparable levels. Procyanidin dimer type B isomer I was similar across all donors. Ellagic acid was most abundant in D3, while D1 and D2 showed similar levels; in contrast, methylellagic acid *O*-pentoside isomer II was highest in D1 and comparable between D2 and D3. Tormentic acid *O*-hexoside and tetrahydroxy-11-methoxy-urs-12-en-28-oic acid-28-*O*-hexoside showed comparable amounts in donors 2 and 3, slightly exceeding those in donor 1. Tormentic acid *O*-hexoside isomer II was clearly dominant in donor 3, while donors 1 and 2 displayed similar, moderately lower levels. Trihydroxy-urs-12-en-28-oic acid-28-*O*-hexoside isomer and dihydroxy-3-oxo-urs-12-en-28-oic acid-28-*O*-hexoside followed the same pattern, being most abundant in donor 3. In contrast, dihydroxy-oxo-urs-12-en-28-oic acid-*O*-hexoside isomer was highest in D2, while cecropiacic acid was slightly higher in donor 3, with donors 1 and 2 showing comparable levels. The remaining minor compounds, including undefined triterpenoic acid hexoside and tormentic acid isomer I, showed a gradual increase from donor 1 to donor 3, whereas tormentic acid isomer II was most abundant in D1.

These interindividual variations likely reflect differences in gut microbiota composition and metabolic activity among the donors.

In the post-incubation mixtures, several new compounds were detected, representing possible gut microbiota-derived metabolites. As shown, only catechin and procyanidin derivatives underwent detectable microbial transformation, whereas the remaining extract constituents were not metabolized under the experimental conditions. As shown in Figure 2, catechin and procyanidin metabolites were selectively detected in both 30% and 100% MeOH fractions, with distinct polarity-dependent distribution. The 30% MeOH fractions contained the highest relative abundances of catechin-type derivatives, consistent with their higher polarity and better solubility in aqueous methanol. In contrast, 100% MeOH fractions showed a relatively greater contribution of procyanidin-type compounds, reflecting their more lipophilic nature. The qualitative metabolite pattern remained consistent across donors (D1–D3), whereas quantitative differences were apparent—donor 3 exhibited the highest overall intensities for both catechin and procyanidin derivatives, suggesting higher microbial metabolic activity.

These results confirm that flavan-3-ols and their oligomers represent the main microbiota-metabolized constituents of the *Tormentillae rhizoma* extract and that both solvent polarity and interindividual microbiota variability modulate their metabolic profiles.

### 2.3. Human Isolated Neutrophils Model

#### 2.3.1. Influence of Extract and Its Gut Metabolites on Cell Viability, Apoptosis and Necrosis

All experiments on immune cells were performed using ultrafiltered extracts to remove high-molecular-weight pyrogens, following the procedure described in a previous study [30]. This step was essential, as the presence of pyrogenic contaminants in plant extracts can induce apparent cytokine stimulation and confound the interpretation of immunomodulatory effects.

The effect of tormentil rhizome ethanolic extract and its gut microbiota–derived metabolites fraction on neutrophil viability and apoptosis was evaluated using Annexin V-FITC/PI staining followed by flow cytometry (Figure 3). Human isolated neutrophils were treated with EtTR or TRGM samples at concentrations ranging from 31.25 to 250 µg/mL and stimulated with LPS (100 ng/mL). The proportion of live, early apoptotic, late apoptotic, and necrotic cells was expressed as a percentage relative to the LPS-stimulated control (st(+)). Roscovitine (Rsc, 50 µM) served as a positive control for apoptosis induction, while non-stimulated cells (st(−)) represented the baseline. Panel A shows the quantitative distribution of neutrophil death modes, whereas Panel B presents representative Annexin V–FITC/PI dot plots illustrating the gating strategy and raw cytometry signals for selected treatment conditions.

Flow cytometric analysis confirmed that treatment with EtTR and TRGMs did not compromise neutrophil viability. Across all tested concentrations (31.25–250 µg/mL), the proportion of viable cells remained consistently high—typically above 90%, comparable to the LPS-stimulated control (94%). Only minor variations were observed at the highest extract concentration (250 µg/mL), where viability reached approximately 88%, accompanied by small increases in late apoptotic (≈4%) and necrotic cells (≈4%). All TRGM fractions, regardless of donor or extraction solvent, maintained viability within 90–97%, with apoptotic and necrotic cells each below 3%. These slight fluctuations were not statistically significant. In contrast, roscovitine, used as a positive control, markedly reduced cell survival and increased apoptosis, confirming the sensitivity of the assay.

Overall, these results indicate that EtTR and its gut microbiota-derived metabolites are non-cytotoxic to neutrophils, preserving cellular integrity and viability even under inflammatory stimulation with LPS.

#### 2.3.2. Influence of Extract and Its Gut Metabolites on ROS Production

The effect of *Tormentillae rhizoma* ethanolic extract and its gut microbiota–derived metabolites on ROS generation in f-MLP-stimulated human neutrophils was evaluated (Figure 4). The results are presented as ROS production (%), representing the level of reactive oxygen species generated by neutrophils expressed as a percentage relative to the f-MLP-stimulated control (st(+)). Quercetin (Qr, 20 µM) was used as a positive antioxidant control.

As expected, exposure of neutrophils to f-MLP (1.5 μg/mL) markedly increased intracellular ROS generation compared with non-stimulated cells. Treatment with tormentil rhizome extract produced a clear, concentration-dependent antioxidant effect, significantly lowering ROS levels (*p* ≤ 0.05 vs. f-MLP control). The most pronounced inhibition was observed at 125–250 µg/mL, where fluorescence values declined to approximately 30–45% of the stimulated control. All gut microbiota-derived metabolite fractions also attenuated oxidative activity. Fractions obtained using 100% methanol exhibited the strongest effect, reducing ROS formation to around 35–50% of control, whereas 30% methanol fractions achieved slightly weaker suppression, typically 55–70% of control. Fractions obtained using 100% methanol generally showed stronger inhibition than 30% methanol extracts, particularly for donors D1 and D2, where ROS levels decreased to around 35–50% of control. In contrast, the D3 MeOH100 fraction displayed a slightly weaker reduction, maintaining ROS levels at approximately 50–60% of control at 31.25 and 250 µg/mL, while showing elevated values (around 110–130%) at 62.5 and 125 µg/mL. Differences between donors were minor overall. The positive control, quercetin, effectively suppressed ROS generation to below 25%, confirming assay reliability.

These findings demonstrate that tormentil rhizome extract and its microbiota-derived metabolites exert potent antioxidant and ROS-suppressing effects in f-MLP-stimulated neutrophils, supporting their potential to alleviate inflammation-associated oxidative stress.

#### 2.3.3. Influence of Extract and Its Gut Metabolites on Cytokine Secretion

The effect of tormentil rhizome ethanolic extract and its gut microbiota–derived metabolites on the secretion of pro-inflammatory cytokines IL-1β, IL-8, and TNF-*α* in LPS-stimulated human neutrophils was evaluated (Figure 5). Cytokine levels are expressed as percentages relative to the LPS-stimulated control (st(+)), and dexamethasone (Dex, 20 µM) was used as a positive anti-inflammatory control.

LPS stimulation (100 ng/mL) strongly increased IL-1*β* secretion compared with non-stimulated cells (st(−)). Treatment with tormentil rhizome extract caused a reduction in IL-1*β* levels, reaching approximately 45–50% of the LPS control at 125 µg/mL and around 55% at 62.5 µg/mL. At the highest concentration (250 µg/mL), EtTR increased IL-1*β* secretion (≈170% of LPS-stimulated control). Overall, TRGM fractions also suppressed IL-1*β* release, though the magnitude of inhibition varied among donors and extraction solvents. The most effective inhibition of IL-1*β* secretion was observed for the TRGM_D1_MeOH30 (65–70% of control at 31.25–125 µg/mL) and TRGM_D2_MeOH100 fractions (65–75% at 31.25–250 µg/mL). A weaker effect was noted for TRGM_D2 MeOH30 (70–75% at 31.25–62.5 µg/mL) and TRGM_D3 MeOH30 (75–80% at 62.5–125 µg/mL). In contrast, the TRGM_D1 MeOH100 fraction showed only a minor reduction (≈81% of control at 62.5 µg/mL), while at other concentrations its effect did not differ significantly from the stimulated control. The TRGM_D3 MeOH100 fraction did not suppress IL-1*β* release at any tested concentration. Dexamethasone strongly inhibited IL-1*β* secretion to below 10%, confirming assay responsiveness.

The pattern for IL-8 release differed markedly. While LPS stimulation substantially elevated IL-8 production, treatment with EtTR and TRGM further increased secretion under most conditions, typically reaching 120–180% of the LPS control. The most pronounced enhancement was observed for extract, TRGM_D1_MeOH100 and both fractions from donor D3 (130–200%), suggesting that certain microbial metabolites may potentiate neutrophil activation and chemokine release. Only at 62.5 µg/mL, the TRGM_D1 MeOH30 fraction reduced IL-8 secretion to approximately 90% of the control, while the TRGM_D2 MeOH100 fraction lowered it to about 80%, indicating donor-dependent variability. As expected, dexamethasone effectively suppressed IL-8 secretion to below 20% of control.

In contrast, TNF-*α* production was consistently downregulated by both EtTR and TRGM treatments. The parent extract reduced TNF-*α* levels to approximately 40–70% of the LPS control, except at the highest concentration, where a marked stimulation to around 200% was observed. Similar or even stronger inhibition (35–45% of control) was noted for the TRGM_D1 MeOH30 and TRGM_D2 MeOH100 fractions, except at 250 µg/mL. A slightly weaker effect was observed for TRGM_D1 MeOH100 and TRGM_D2 MeOH30 (except at 250 µg/mL, where values did not differ significantly from the LPS control), as well as for TRGM_D3 MeOH30 (45–65% of control). The TRGM_D3 MeOH100 fraction displayed a more variable response, showing inhibition (55–75% of control) at lower concentrations but a mild increase (≈110%) at 250 µg/mL. Dexamethasone markedly reduced TNF-*α* secretion to about 8% of the control.

Overall, these results demonstrate that tormentil rhizome extract and its gut microbiota-derived metabolites exert differential immunomodulatory effects on neutrophils, strongly suppressing IL-1*β* and TNF-*α* while enhancing IL-8 secretion.

### 2.4. THP-1 Macrophages Model

#### 2.4.1. Influence of Extract and Its Gut Metabolites on Cell Viability

The effect of tormentil rhizome ethanolic extract and its gut microbiota-derived metabolites on the viability of LPS-stimulated THP-1 macrophages was assessed using the MTT assay (Figure 6). Cell viability is expressed as a percentage relative to the LPS-stimulated control (st(+)). Triton X-100 (Trt, 0.1% *v*/*v*) was used as a positive cytotoxic control.

Exposure of THP-1 macrophages to LPS (100 ng/mL) did not cause any significant reduction in metabolic activity compared with unstimulated cells, which exhibited slightly higher viability (approximately 130% of the LPS control). Treatment with tormentil rhizome extract across the tested concentration range (31.25–250 µg/mL) did not affect cell viability, maintaining values between 95 and 115% of the LPS control. Similarly, all gut-derived metabolite fractions (TRGM_D1–D3) obtained using 30% and 100% methanol were non-cytotoxic, with live-cell percentages typically ranging from 90 to 115%. Minor variations between donors or solvent systems were not statistically significant. In contrast, Triton X-100 drastically reduced viability to below 10%, validating the assay’s performance and sensitivity.

These findings confirm that both EtTR extract and its microbiota-derived metabolites are safe for THP-1 macrophages, even under inflammatory stimulation, and can be reliably used in subsequent functional and mechanistic studies.

#### 2.4.2. Influence of Extract and Its Gut Metabolites on Cytokine Secretion

The effect of EtTR ethanolic extract and its gut microbiota-derived metabolites on the secretion of pro-inflammatory cytokines IL-6, MCP-1, and TNF-*α* in LPS-stimulated THP-1 macrophages was evaluated (Figure 7). Cytokine levels are expressed as percentages relative to the LPS-stimulated control (st(+)). Dexamethasone (Dex, 20 µM) served as a positive anti-inflammatory control. Additionally, the secretion of the anti-inflammatory cytokine IL-10 was analyzed; however, its levels were below the lowest point of the standard curve, and therefore, the results are not presented.

Stimulation of THP-1 macrophages with LPS (100 ng/mL) strongly increased IL-6 secretion compared with non-stimulated cells. Treatment with tormentil rhizome extract reduced IL-6 release only at the lowest concentration (≈80% of control), while higher concentrations produced levels similar to the LPS-stimulated control. Among the microbiota-derived fractions, the effects were clearly donor-dependent. The TRGM_D1_MeOH30 fraction (except at 125 µg/mL, ≈87%) and the TRGM_D2_MeOH100 fraction (except at 31.25 µg/mL, ≈120%) maintained IL-6 close to control values. In contrast, the TRGM_D2 100% and TRGM_D3 30% fractions markedly enhanced IL-6 secretion, reaching 160–200% of the LPS control, while TRGM_D1_100% increased it to around 150%. The TRGM_D3 100% fraction showed a moderate stimulatory trend at higher concentrations, whereas at the lowest concentration, IL-6 remained comparable to the stimulated control. As expected, dexamethasone strongly suppressed IL-6 secretion to below 30% of the control, confirming the responsiveness of the assay.

LPS stimulation also led to a pronounced increase in MCP-1 secretion. Treatment with EtTR decreased MCP-1 production to about 80% of the LPS control at the lowest concentrations, while higher concentrations did not differ significantly from the stimulated control. An inhibitory trend was observed for the TRGM_D1 MeOH30 fraction at 62.5 and 250 µg/mL, reducing MCP-1 to approximately 90% and 75% of control, respectively, and for TRGM_D3, where the MeOH30 fraction at the highest concentration decreased MCP-1 to about 90%, and the MeOH100 fraction at the lowest concentration to about 85%. In contrast, the TRGM_D2 fractions induced MCP-1 secretion, occasionally exceeding 130–165% of control values (MeOH30 at 31.25 µg/mL and MeOH100 at 62.5–125 µg/mL), indicating donor-specific variation in the inflammatory response. The remaining samples did not differ significantly from the LPS-stimulated control. Dexamethasone robustly reduced MCP-1 secretion, confirming assay sensitivity.

TNF-*α* production was consistently downregulated by both EtTR and most TRGM samples. The parent extract at concentrations of 31.25–62.5 µg/mL decreased TNF-*α* levels to approximately 80–85% of the LPS control. The TRGM_D1 fraction reduced TNF-*α* secretion only at the lowest concentration of MeOH100 (≈80%), while both TRGM_D2 fractions showed inhibition at the highest concentration (≈70% and 85% of control). TRGM_D3 displayed a variable pattern—showing stimulation in the MeOH30 fraction (130–180% of control), but a slight decrease in the MeOH100 fraction (65–90%) at 31.25–62.5 µg/mL. Other samples did not differ significantly from the control. Dexamethasone almost completely abolished TNF-*α* secretion, reducing it to below 20% of the control.

Collectively, these results demonstrate that tormentil rhizome extract and its microbiota-derived metabolites modulate macrophage cytokine secretion in a concentration- and donor-dependent manner, showing both inhibitory and stimulatory effects that reflect complex immunomodulatory activity.

## 3. Discussion

Because neutrophils and macrophages play distinct but complementary roles in the inflammatory response, the experimental design and readouts were adapted to their biological characteristics. Neutrophils are short-lived, non-adherent immune cells that respond rapidly to pro-inflammatory stimuli through oxidative burst and cytokine release [21,22,23]. Therefore, their functional activity was evaluated by assessing cell viability and death mode (apoptosis/necrosis), ROS generation, and the secretion of IL-8, IL-1*β*, and TNF-*α*, which are key markers of acute neutrophil activation. In contrast, THP-1-derived macrophages are adherent, metabolically active cells that represent a model of sustained inflammatory signaling [27]. Accordingly, cytotoxicity was determined using the MTT assay, reflecting mitochondrial activity, while cytokine secretion was analyzed for MCP-1, IL-6, and TNF-*α*—typical mediators of macrophage-driven inflammatory and chemotactic responses. This complementary experimental approach enabled the evaluation of both early oxidative and late cytokine-mediated events in innate immune activation, providing a comprehensive overview of the immunomodulatory potential of tormentil rhizome extract and its microbial metabolites.

The biological activity of the tormentil rhizome extract and its gut microbiota–derived metabolites reflected their distinct chemical composition. The ethanolic extract, dominated by catechins, procyanidins, and triterpenoid acids, exhibited strong antioxidant and anti-inflammatory properties, consistent with the high abundance of redox-active polyphenols and membrane-stabilizing triterpenes [5,10,11,12]. Its ability to reduce ROS generation and suppress IL-1*β* and TNF-*α* secretion in LPS-stimulated neutrophils indicates direct modulation of early inflammatory signaling, likely through inhibition of NADPH oxidase-dependent oxidative burst and attenuation of NF-κB activation. Similar antioxidant and anti-inflammatory effects of catechin- and procyanidin-rich fractions were previously reported for other *Potentilla* species, including *P. supina* and *P. paradoxa*, where ethanol extracts reduced NO and PGE_2_ production and suppressed iNOS and COX-2 expression in LPS-stimulated RAW 264.7 macrophages through NF-κB and AP-1 inhibition. Among the metabolite fractions, biological effects corresponded closely to their chemical profiles. Fractions eluted with 30% MeOH were enriched in catechins and low-molecular-weight procyanidins, whereas the 100% MeOH fractions contained mainly ellagic acid derivatives and glycosylated triterpenoids such as tormentic acid *O*-hexoside and its isomers. The strongest suppression of ROS and pro-inflammatory cytokines (IL-1*β* and TNF-*α*) was observed for TRGM_D1_MeOH30 and TRGM_D2_MeOH100, aligning with their high content of catechin-type polyphenols and triterpenoid conjugates. In contrast, fractions poorer in these constituents (e.g., D1_MeOH100 and D3_MeOH30) showed weaker or inconsistent effects, whereas D3_MeOH100, enriched in triterpenoids but containing fewer catechins, displayed a biphasic response with mild stimulation at higher concentrations. These findings suggest that both phenolic and triterpenoid components contribute to the anti-inflammatory action, but their relative ratio and polarity determine the overall biological outcome. The stimulation of IL-8 secretion observed for several samples, particularly at higher concentrations, may reflect a context-dependent, redox-linked chemokine response rather than a straightforward pro-inflammatory effect. Upregulation of IL-8 through EGFR/NF-κB-dependent modulation by polyphenols, including resveratrol, has been documented in epithelial models (HaCaT and NHEK cells stimulated with TNF-*α*), whereas in innate immune systems, the response appears more variable and depends on the compound type, dose, and inflammatory stimulus [31,32]. In innate immune cells, however, increased IL-8 secretion has a direct functional consequence, as IL-8 is one of the major chemokines responsible for neutrophil recruitment and activation. Even moderate elevations of IL-8 can enhance neutrophil chemotaxis, promote firm adhesion to the endothelium, and prime these cells for oxidative burst and degranulation. Therefore, the IL-8 increase observed in our neutrophil model may reflect an adaptive chemotactic response aimed at improving early immune surveillance rather than a purely pro-inflammatory effect. Such controlled IL-8 upregulation is consistent with a transient, redox-linked signaling shift that prepares neutrophils for environmental stress without inducing excessive inflammatory damage [33,34].

In THP-1 macrophages, the tormentil extract and its gut microbiota-derived fractions modulated cytokine secretion in a donor-dependent manner, displaying both inhibitory and stimulatory effects. The ethanolic extract reduced TNF-*α* and, to a lesser extent, MCP-1 levels at lower concentrations, which aligns with the suppression of NF-κB-dependent transcription observed for other *Potentilla* species [13,14,15]. In contrast, IL-6 secretion remained largely unaffected or was even enhanced at higher concentrations, suggesting a shift from inhibitory to adaptive signaling, possibly linked to moderate redox activation or altered feedback regulation of cytokine expression. Among the microbial metabolite fractions, TRGM_D1_MeOH30 and TRGM_D2_MeOH100 showed the most pronounced inhibitory effects, reducing TNF-*α* and, in part, MCP-1 secretion, consistent with their enrichment in catechins, ellagic acid derivatives, and triterpenoid glycosides. In contrast, fractions with lower phenolic content, such as TRGM_D2_MeOH30 and TRGM_D3_MeOH30, tended to enhance IL-6 and MCP-1 release, indicating donor-dependent differences in the metabolic transformation of tormentil constituents. Such divergence likely reflects variation in microbial enzymatic capacity and metabolite composition, as previously observed for ellagitannin-derived urolithins, which exhibit distinct immunomodulatory properties depending on their structure and conjugation state [16,17,18,19,20]. In macrophage models, urolithin A has been shown to attenuate TNF-*α* and enhance IL-10 via ERK1/2 signaling, whereas its glucuronide conjugates remain inactive [16]. These findings support the view that microbial transformation of tormentil polyphenols generates low-molecular-weight metabolites capable of fine-tuning macrophage cytokine responses rather than producing uniform suppression.

Taken together, the results obtained from both neutrophil and macrophage models demonstrate that *Tormentillae rhizoma* extract and its microbiota-derived metabolites exert complex, concentration- and donor-dependent immunomodulatory effects. The consistent inhibition of oxidative burst and pro-inflammatory cytokines such as IL-1*β*, TNF-*α*, and MCP-1 reflects the strong antioxidant and anti-inflammatory potential of tormentil constituents, particularly flavan-3-ols and triterpenoid acids [5,10,11,12,13,14,15]. At the same time, the variable modulation of IL-6 and IL-8 secretion indicates a regulatory rather than purely suppressive mode of action, which may contribute to restoring immune homeostasis under inflammatory stress.

The interplay between extract composition, solvent polarity, and microbial metabolism appears to be a key determinant of the observed bioactivity. The interindividual variability among donors highlights the crucial role of the gut microbiota in shaping the metabolic and immunological outcomes of polyphenol intake [1,2,3]. These findings support the emerging concept that hydrolyzable tannins from tormentil rhizome act as precursors of bioactive postbiotic metabolites capable of fine-tuning innate immune responses rather than exerting uniform suppression. Collectively, the study provides mechanistic evidence that tormentil rhizome and its microbiota-derived metabolites can modulate oxidative and inflammatory pathways in innate immune cells through complementary redox and signaling mechanisms. Such properties underline the potential of tormentil rhizome as a source of functional ingredients for the prevention or management of inflammation-related intestinal and systemic disorders.

## 4. Materials and Methods

### 4.1. Materials

Plant material, tormetil rhizome, with serial No. 946.2021 was purchased from Kawon (Gostyń, Poland). All solvents, including dimethyl sulfoxide, ethanol and methanol, were purchased from Avantor (Gliwice, Poland). LC-MS grade solvents, such as water and acetonitrile, were purchased from Witko (Łódź, Poland). Ammonium formate and formic acid of LC-MS grade were obtained from Chem-lab (Zedelgem, Belgium). Brain heart infusion (BHI) broth was purchased from bioMerieux SA (Craponne, France). Fetal bovine serum (FBS), phosphate-buffered saline (PBS), RPMI 1640 with stable glutamate and 25 mM of HEPES with and without phenol red, penicillin-streptomycin solution and Hanks’ Balanced Salt Solution (HBSS) without calcium and magnesium were obtained from Biowest (Nuaillé, France). Thiazolyl blue tetrazolium bromide (MTT) was sourced from Acros Organics B.V.B.A. (Geel, Belgium). Lipopolysaccharide (LPS) from *E. coli* O111:B4 and Triton X-100 (Trt) were purchased from Merck Life Science (Darmstadt, Germany). Dextran from *Leuconostoc mesenteroides*, dexamethasone (Dex), quercetin (Qr), formyl-methionyl-leucyl-phenylalanine (f-MLP), luminol, phorbol-12-myristate-13-acetate (PMA), roscovitine (Rsc), and propidium iodide (PI) were obtained from Sigma-Aldrich (St. Louis, MO, USA). Annexin V–fluorescein isothiocyanate (Annexin V–FITC), binding buffer, and ELISA sets were purchased from BD (Franklin Lakes, NJ, USA). Pancoll was purchased from Biotech (Aidenbach, Germany). Ultra-pure water was prepared using a Simplicity UV system from Merck Millipore.

### 4.2. Preparation of Plant Extract

The ethanolic extract of tormentil rhizome was obtained by a three-step extraction using a mixture of ethanol and water (7:3, *v*/*v*). At each step, the plant material was extracted with 500 mL of the solvent, supported by ultrasonic treatment in a water bath for 30 min at 40 °C. The obtained extract was concentrated using a rotary evaporator (240 mbar, 35 °C water bath, flask rotation speed 120 rpm) to remove ethanol, and subsequently lyophilized to remove water (48 h, vacuum <1 mbar, collector temperature −80 °C). The yield of the dry extract after lyophilization was 13.6%. The extract was stored in a sealed container at −17 °C [6,28].

### 4.3. Quantitative Analysis of the Extract

Quantitative analysis of the ethanolic extract of *Tormentillae rhizoma* was performed using a validated UHPLC-DAD-MS/MS method as previously described [6]. Calibration curves were prepared for representative analytical standards, including (–)-epicatechin, agrimoniin, *α*-hederin, and 3,3′-di-*O*-methylellagic acid 4′-xylopyranoside. The method demonstrated excellent linearity (*r* > 0.997), with limits of detection (LOD) and quantification (LOQ) ranging from 0.05 to 3.13 ng and 0.16 to 939 ng per injection, respectively. Intra- and interday precision values were below 5%. Quantification was based on external calibration and peak area integration at compound-specific wavelengths using DAD and MS data for confirmation. The results were expressed as the percentage contribution of each compound to the total amount of quantified constituents in the extract [6].

### 4.4. Biosynthesis of Tormentil Rhizome Gut Metabolites

The study was approved by the Ethical Committee of the Medical University of Warsaw (approval no. AKBE/151/2021) on 6 September 2021 and conducted in accordance with the Declaration of Helsinki. Fecal samples were collected from three healthy adult donors (one woman and two men, aged 26–36) who met the inclusion criteria, ensuring sample quality and safety, including no history of gastrointestinal or infectious diseases, no antibiotic use within the previous six months, and adherence to a phenolic compound-restricted diet for three days before donation. Fecal samples were processed within 30 min after collection to prepare fecal slurries (FS; 1:10 *m*/*v* in liquid brain heart infusion, 37 °C). For each donor, 3 mL of FS was mixed with 12.5 mL of tormentil rhizome ethanolic extract and 234.5 mL of brain heart infusion medium. Incubations were carried out under anaerobic conditions (Bactron 300, Sheldon Manufacturing, Cornelius, OR, USA) at 37 °C for 24 h. Reactions were terminated at 0 h and 24 h by adding methanol containing 0.1% formic acid (1:1, *v*/*v*). Controls included an extract without FS and blanks containing only FS in the medium. After incubation, samples were centrifuged and subjected to solid-phase extraction (SPE) using 30% and 100% methanol in water. These solvent proportions were selected based on prior experimental optimization, as they efficiently recover a broad range of metabolites: 30% methanol favors the extraction of polar compounds, whereas 100% methanol extracts less polar, lipophilic constituents. The obtained metabolite fractions were concentrated, freeze-dried, and prior to chromatographic analyses, reconstituted, centrifuged, and filtered. Each fraction was assigned a specific code to indicate its origin and extraction conditions: TRGM_D1–D3_MeOH30 and TRGM_D1–D3_MeOH100, where TRGM refers to Tormentil Rhizome Gut Metabolites, D1–D3 denotes the donor number, and MeOH30 or MeOH100 indicates elution with 30% or 100% methanol during SPE [28].

### 4.5. Chromatographic Analysis of Extract

The chemical profiling of tormentil rhizome ethanolic extract was performed on a UHPLC-3000 RS system (Dionex, Leipzig, Germany) coupled to an AmaZon SL ion trap mass spectrometer equipped with an electrospray ionization (ESI) source (Bruker Daltonik GmbH, Bremen, Germany) and a diode array detector (DAD). The system operated in splitless mode. UV spectra were recorded within the 200–450 nm range. The ion source parameters were as follows: nebulizer gas pressure, 40 psi; drying gas (nitrogen) flow rate, 9 L/min; gas temperature, 134 °C; and capillary voltage, 4.5 kV. The instrument scanned ions in the *m*/*z* range of 70–2200. Chromatographic separation was carried out on a Kinetex XB-C18 column (150 mm × 3.0 mm, 2.6 µm; Phenomenex, Torrance, CA, USA). The mobile phases consisted of water with 0.1% formic acid (A) and acetonitrile with 0.1% formic acid (B), both of analytical or LC-MS grade. The elution was performed with a linear gradient from 5 to 26% B over 0–60 min, followed by 26 to 60% B over 60–90 min, at a constant flow rate of 0.3 mL/min. The column temperature was maintained at 25 °C. Prior to injection, samples were passed through a 0.45 µm syringe filter, and an injection volume of 4 µL was used for all analyses [6,28].

### 4.6. Chromatographic Analysis of Tormentil Gut Metabolites

Metabolite profiling of control and experimental samples was conducted using a Vanquish UHPLC system coupled to an Orbitrap Exploris 120 mass spectrometer (Thermo Fisher Scientific, Bremen, Germany). Chromatographic separation was achieved on a Kinetex XB-C18 column (150 × 2.1 mm, 1.7 µm particle size). The mobile phases consisted of (A) water and (B) acetonitrile:water (4:1, *v*/*v*), both containing 0.1% formic acid and 5 mM ammonium formate (NH_4_HCOO). The gradient program was as follows: 0–3.5 min, 1% B; 3.5–16.5 min, 1–26% B; 16.5–26.5 min, 26–100% B; 26.5–28.5 min, 100% B. The flow rate was set at 0.3 mL min^−1^, and the column temperature was maintained at 45 °C. The mass spectrometer was operated in electrospray ionization (ESI) mode with polarity switching. The spray voltage was 3.5 kV in positive and 2.0 kV in negative mode. The sheath, auxiliary, and sweep gas flows were 48, 11, and 2 arbitrary units, respectively. The ion transfer tube and vaporizer temperatures were set to 320 °C and 280 °C, respectively. Full MS data were acquired at a resolving power of 120,000, followed by data-dependent MS/MS (ddMS^2^) scans collected at 15,000 resolution using stepped normalized collision energies (NCE) of 30 and 50%. Four fragmentation scans were triggered per full MS cycle, with a single-charge filter, isotope pattern recognition, and dynamic exclusion enabled. To improve metabolite coverage, experimental samples were analyzed both with and without exclusion lists generated from control runs. All raw data were processed using Compound Discoverer 3.4 (Thermo Fisher Scientific, Austin, TX, USA) for peak alignment, compound annotation, and statistical evaluation [28].

### 4.7. Ultrafiltration of Extract and Its Gut Metabolites

To remove high-molecular-weight pyrogens, all samples underwent ultrafiltration using Microsep™ Advance Centrifugal Devices (Pall Corporation, Port Washington, NY, USA). The extracts were centrifuged sequentially through filters with 100 kDa and 30 kDa molecular weight cut-offs. Subsequently, the filtrates were sterilized by passage through 0.22 µm syringe filters to ensure sterility prior to further assays [30,35].

### 4.8. Isolation of Human Neutrophils

Human neutrophils were obtained from buffy coats provided by the Warsaw Blood Donation Centre. Samples were collected from three healthy male donors under 35 years of age who declared that they were non-smokers and were not taking any medication. Only anonymized buffy coats from donors who had given consent for scientific use of their blood components were used. All donors underwent routine laboratory testing, and only samples with parameters within normal physiological ranges were included. The study was conducted in accordance with the Declaration of Helsinki and approved by the Ethics Committee of the Medical University of Warsaw (AKBE/31/2023) on 16 January 2023. Neutrophils were separated by dextran sedimentation followed by centrifugation using a Pancoll density gradient (1.077 g/mL; 1500 rpm, 4 °C). Residual erythrocytes were removed by hypotonic lysis. The final neutrophil preparations showed a purity exceeding 97%. Cells were suspended in RPMI 1640 medium with phenol red, stable glutamine, 25 mM HEPES, and supplemented 10% FBS and 1% penicillin–streptomycin medium and stored at 4 °C until further use [36,37].

### 4.9. Incubation of Neutrophils with Plant Extracts

The cell suspension from point 4.6 (2 × 10^6^ cells/mL) was distributed into 96-well plates, and the tested EtTR extract and its gut metabolites were added to achieve final concentrations of 31.125, 62.5, 125, or 250 µg/mL. The concentration range was selected based on previous studies using structurally similar polyphenol-rich plant extracts, in which higher doses frequently caused non-specific cytotoxic effects in immune cell models [30,37]. The medium containing DMSO was added to the stimulated and non-stimulated controls (final concentration in well 5%). After 1 h preincubation with the tested samples, neutrophils were stimulated with LPS (100 ng/mL). The cells were then incubated for 24 h at 37 °C in a humidified atmosphere containing 5% CO_2_ [36,37].

### 4.10. Neutrophil Viability, Apoptosis and Necrosis Assay

Cell viability, apoptosis and necrosis were evaluated using Annexin V-FITC and propidium iodide staining. Neutrophils from point 4.6 after 24 h of incubation, cells were centrifuged (2000 rpm, 10 min, 4 °C), supernatants were collected, and the pellets were washed twice with PBS. Neutrophils were then resuspended in binding buffer and transferred to a new 96-well plate at a concentration of 0.5 × 10^6^ cells/well (4-fold dilution). A staining solution containing Annexin V-FITC (5 µL/mL) and PI (0.5 µg/mL) in binding buffer was added to each well. After 15 min of incubation in the dark at room temperature, the cells were analyzed by flow cytometry (BD FACSCelesta, Becton Dickinson, Franklin Lakes, NJ, USA) with laser settings: FSC: 550 V, SSC: 284 V, FITC: 421 V, Per CP-Cy5-5: 601 V. Data were collected from 10,000 gated events. Instrument compensation and quadrant settings were adjusted before each measurement. Roscovitine (50 µM) was added to cells at the same time as the tested samples served as positive controls for apoptosis induction. The cytotoxicity effect of the extract and its metabolites on human neutrophils was expressed as a percentage relative to the stimulated control (100%) [30].

### 4.11. Determination of ROS Production

The production of reactive oxygen species (ROS) was evaluated in isolated human neutrophils stimulated with f-MLP. A volume of 50 μL of the extract or its gut metabolites, dissolved in HBSS at concentrations of 31.25, 62.5, 125, and 250 μg/mL, was placed in white 96-well plates. Subsequently, 70 μL of the neutrophil suspension (3 × 10^6^ cells/mL) was added to each well, followed by 50 μL of luminol (0.4 mg/mL). The reaction was initiated by adding 30 μL of f-MLP (1.5 μg/mL) to the mixture of cells with the tested samples and the stimulated control (st(+)), whereas 30 μL of HBSS was added to the non-stimulated control (st(−)). All samples and reagents were prepared in HBSS without calcium and magnesium ions. Luminescence was recorded immediately after stimulation for 30 min at 2 min intervals. The maximum chemiluminescence value (observed between 10 and 20 min) was used for calculations. The inhibitory effect of the extract and its metabolites on ROS production was expressed as a percentage relative to the stimulated control (100%) [38].

### 4.12. THP-1 Cell Line Culture

THP-1 human monocytic cells (DSMZ, Braunschweig, Germany) were cultured in 75 cm^2^ culture flasks in RPMI 1640 medium without phenol red, containing stable glutamine, 25 mM HEPES, and supplemented with 10% (*v*/*v*) FBS, and 1% (*v*/*v*) penicillin–streptomycin solution. The culture medium was replaced with fresh medium three times per week. Cell confluency was monitored microscopically, and passaging was performed at 90–100% confluency. All cultures and experiments were maintained at 37 °C in a humidified atmosphere with 5% CO_2_ [39].

### 4.13. Incubation of THP-1 Macrophages with Plant Extracts

Before performing the viability and anti-inflammatory assays, monocytes were differentiated into THP-1-derived macrophages. THP-1 monocytes were seeded in 24-well plates at a density of 4 × 10^5^ cells per well and cultured at 37 °C in a humidified atmosphere containing 5% CO_2_. The culture medium consisted of RPMI 1640 without phenol red, supplemented with 10% FBS and 2 mM glutamine. To induce differentiation into macrophages, cells were treated with 25 ng/mL PMA for 48 h, followed by medium replacement and an additional 24 h resting period. Successful differentiation was confirmed by characteristic macrophage features, including cell adherence, morphological changes, and responsiveness to bacterial LPS stimulation. After the resting period, the tested tormentil rhizome extract and its gut metabolites were added to the cells to obtain final concentrations of 31.25, 62.5, 125, and 250 µg/mL. The same concentration range was used as in neutrophil assays, as it had previously been shown to be non-cytotoxic for polyphenol-rich plant extracts [30,37]. For both the stimulated and non-stimulated controls, culture medium containing DMSO was added to reach a final DMSO concentration of 0.5%. Following the addition of the test samples, the cells were incubated for 1 h, after which LPS (100 ng/mL) was added to the appropriate wells to induce inflammation [39].

### 4.14. Determination of Cytokine Secretion

Supernatants from points were collected after 24 h incubation from point 4.6 and were analyzed for cytokine production (IL-8, TNF-*α*, and IL-1*β*) using commercial Enzyme-Linked Immunosorbent Assay (ELISA) kits following the manufacturer’s instructions. Absorbance was measured at 450 nm with background correction at 570 nm using a BioTek microplate reader. Dexamethasone (20 µM) added to cells at the same time as the tested samples was used as a positive control. The modulatory effect of the extract and its metabolites on cytokine production was expressed as a percentage relative to the stimulated control (100%).

### 4.15. MTT Viability Assay

Cell viability was evaluated using the MTT assay. THP-1-derived macrophages stimulated with LPS, after 24 h incubation with the tested samples described in Section 4.12, were washed twice with warm PBS. After washing, MTT solution (0.5 mg/mL in culture medium) was added to each well. Following 4 h of incubation, the medium containing MTT was removed, and the resulting formazan crystals were dissolved in DMSO. A 0.1% (*v*/*v*) Triton X-100 solution in culture medium was used as a positive control. Absorbance at λ = 570 nm with the correction to 630 nm was measured using a microplate reader (Synergy 4, BioTek, Winooski, VT, USA) [6].

### 4.16. Statistical Analysis

Data processing and statistical evaluations were carried out using Microsoft Excel 2016 (Microsoft Corp., Redmond, WA, USA) and GraphPad Prism 8 (GraphPad Software, San Diego, CA, USA). The results are expressed as mean values ± standard deviation (SD) obtained from at least three independent experiments. Differences between groups were analyzed using one-way analysis of variance (ANOVA) followed by Dunnett’s multiple comparison test. A *p*-value below 0.05 was considered statistically significant.

## 5. Conclusions

This study demonstrates that the ethanolic extract of *Tormentillae rhizoma* and its gut microbiota-derived metabolites exert complementary antioxidant and immunomodulatory effects in human innate immune cells. In neutrophils, EtTR and selected TRGMs significantly reduced ROS as well as IL-1*β* and TNF-*α*, while IL-8 exhibited moderate induction. In THP-1 macrophages, EtTR suppressed TNF-*α* and MCP-1 but variably affected IL-6, indicating donor-dependent modulation. The strongest inhibition was observed for fractions enriched in catechins and triterpenoid conjugates, consistent with synergistic activity of these compound classes. The observed activity pattern suggests that microbial transformation of tormentil compounds generates bioactive postbiotic metabolites capable of fine-tuning inflammatory signaling rather than inducing uniform suppression. These findings provide mechanistic insight into the health-promoting potential of tormentil rhizome and support its further exploration as a functional ingredient targeting oxidative and inflammatory stress.

## Figures and Tables

**Figure 1 ijms-26-11164-f001:**
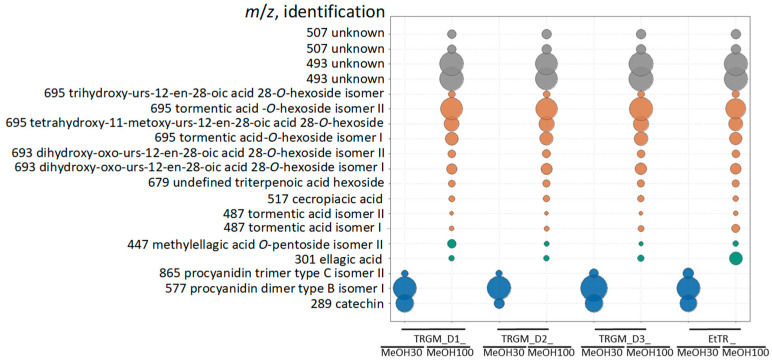
EtTR constituents detected in TRGMs fractions. Compound classes: blue—flavan-3-ols and their oligomers, green—ellagic acid derivatives, orange—triterpenoids and derivatives, grey—unknown compounds. EtTR fractions containing only the eluted extract without fecal slurries were used as a control. Bubble size reflects the relative content (peak area) of each compound. Compound identification was based on previously published LC–MS/MS data [28].

**Figure 2 ijms-26-11164-f002:**
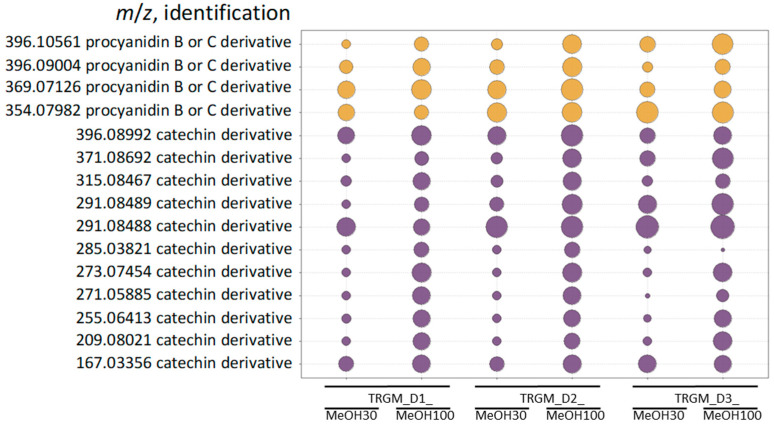
Possible tormentil rhizome gut metabolites detected in TRGMs fractions. Compound classes: violet—catechin derivatives, yellow—procyanidin derivatives. Bubble size reflects the relative content (peak area) of each compound. Compound identification was based on previously published LC–MS/MS data [28].

**Figure 3 ijms-26-11164-f003:**
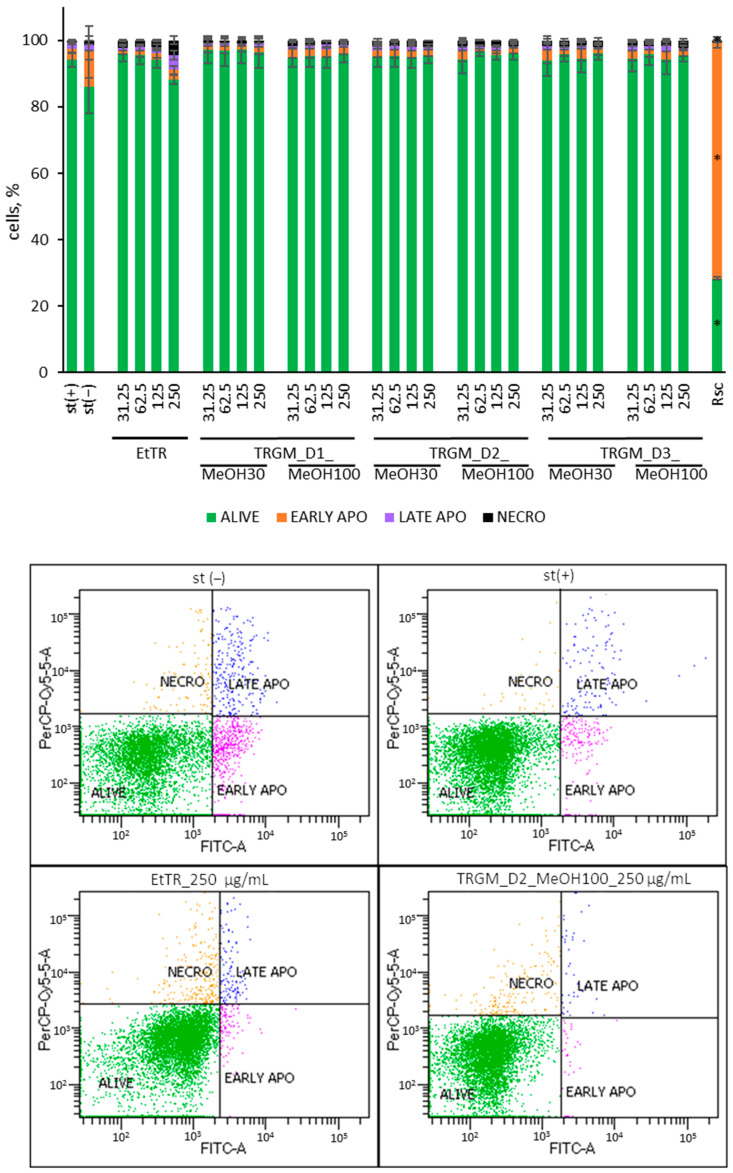
Effect of EtTR and TRGM (31.25–250 µg/mL) on viability, apoptosis and necrosis of isolated human neutrophils stimulated with LPS (100 ng/mL). Upper panel: quantitative distribution of viable, early apoptotic, late apoptotic, and necrotic cells expressed as a percentage of the LPS-stimulated control (st(+)). As a positive control, roscovitine (Rsc, 50 μM) was used. Data were expressed as mean ± SD of three separate experiments conducted in triplicate. Statistical significance was determined by Dunnett’s post hoc test at *p* ≤ 0.05 versus st(+) (*). Lower panel: representative Annexin V–FITC/PI dot plots illustrating the gating strategy and distribution of neutrophil populations (alive, early apoptotic, late apoptotic, necrotic) for selected treatment conditions.

**Figure 4 ijms-26-11164-f004:**
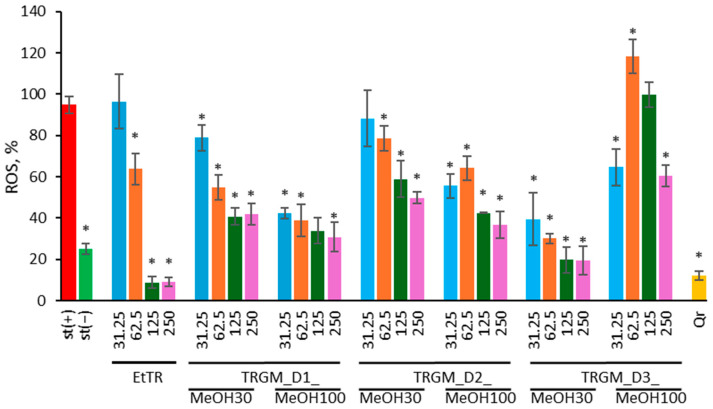
Effect of EtTR and TRGM (31.25–250 µg/mL) on ROS production by f-MLP (1.5 μg/mL) stimulated neutrophils. The ROS level is expressed as a percentage relative to the f-MLP-stimulated control (st(+)). As a positive control, quercetin (Qr, 20 μM) was used. Data were expressed as mean ± SD of three separate experiments conducted in triplicate. Statistical significance was determined by Dunnett’s post hoc test at *p* ≤ 0.05 versus st(+) (*).

**Figure 5 ijms-26-11164-f005:**
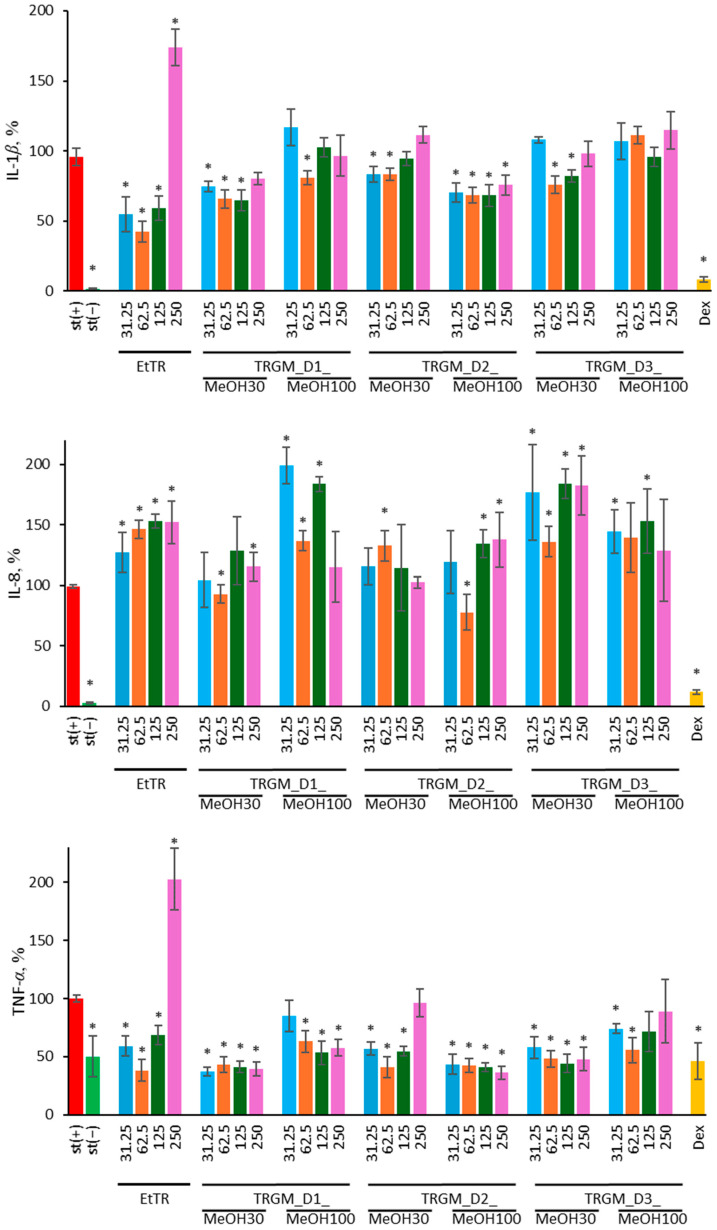
Effect of EtTR and TRGMs (31.25–250 µg/mL) on IL-8, IL-1*β* and TNF-*α* secretion in isolated human neutrophils stimulated with LPS (100 ng/mL). Interleukin levels are expressed as a percentage relative to the LPS-stimulated control (st(+)). As a positive control, dexamethasone (Dex, 20 µM) was used. Data were expressed as mean ± SD of three separate experiments conducted in triplicate. Statistical significance was determined by Dunnett’s post hoc test at *p* ≤ 0.05 versus st(+) (*).

**Figure 6 ijms-26-11164-f006:**
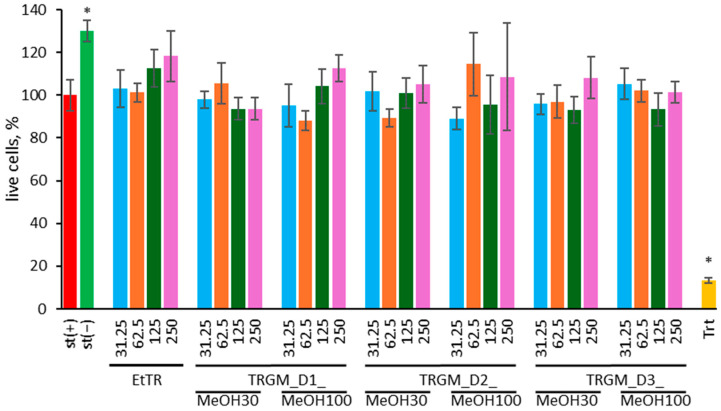
Effect of EtTR and TRGM (31.25–250 µg/mL) on the viability of THP-1 macrophages stimulated with LPS (100 ng/mL). The proportion of cells is expressed as a percentage relative to the LPS-stimulated control (st(+)). As a positive control, Triton X-100 (Trt, 0.1%, *v*/*v*) was used. Data were expressed as mean ± SD of three separate experiments conducted in triplicate. Statistical significance was determined by Dunnett’s post hoc test at *p* ≤ 0.05 versus st(+) (*).

**Figure 7 ijms-26-11164-f007:**
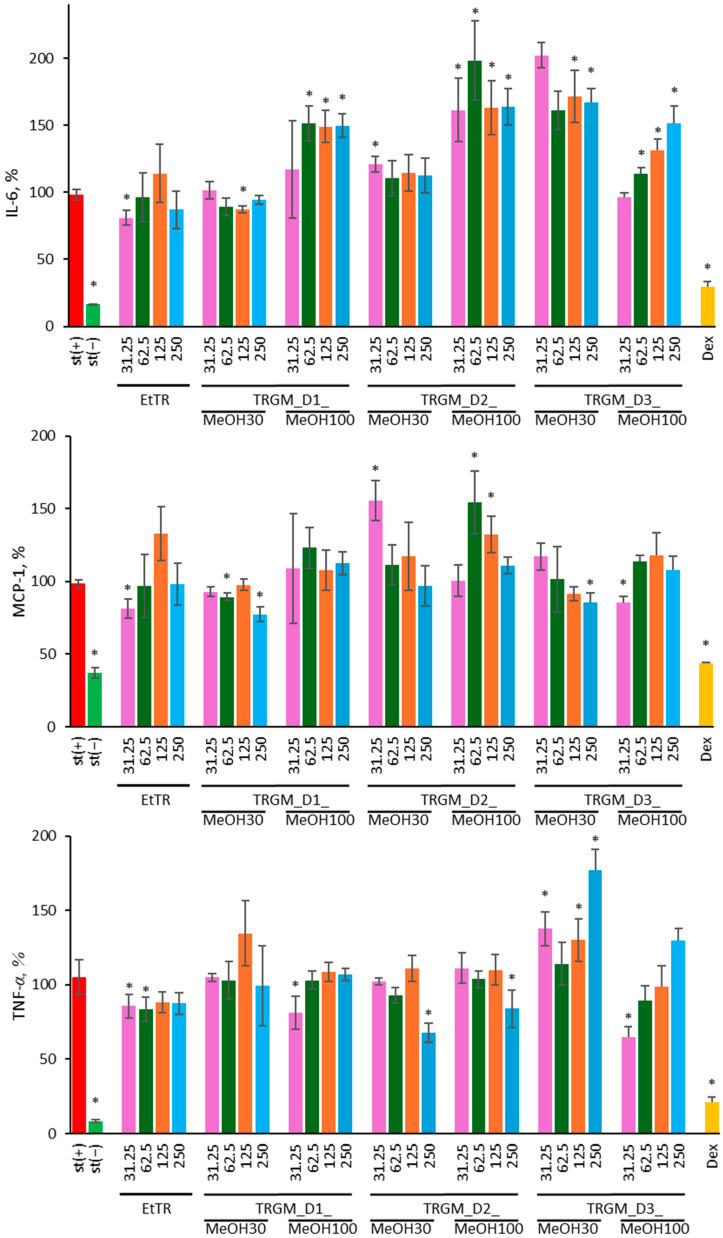
Effect of EtTR and TRGMs (31.25–250 µg/mL) on IL-6, MCP-1 and TNF-*α* secretion in THP-1 macrophages stimulated with LPS (100 ng/mL). Interleukin levels are expressed as a percentage relative to the LPS-stimulated control (st(+)). As a positive control, dexamethasone (Dex, 20 µM) was used. Data were expressed as mean ± SD of three separate experiments conducted in triplicate. Statistical significance was determined by Dunnett’s post hoc test at *p* ≤ 0.05 versus st(+) (*).

**Table 1 ijms-26-11164-t001:** Summary of compound classes identified in the tormentil rhizome ethanolic extract by LC–MS/MS, including representative compounds, diagnostic fragment ions, and quantitative content. Both qualitative identification and quantitative data were based on previously published LC–MS/MS analyses [6,28].

Class	Identification	*m*/*z*	Characteristic MS2 Fragments (*m*/*z*)	Content, % of Determined Compounds
Flavan-3-ols and their oligomers	catechin	289	289, 245, 425, 577	68.0
	catechin-*O*-hexoside	451		
	catechin and (epi)afzelechin dimer	561		
	procyanidin dimer type B isomer I–IV	577		
	procyanidin tetramer isomer I–V	576		
	procyanidin pentamer isomer I–III	720		
	procyanidin trimer type C isomer I–III	865		
Ellagic acid derivatives and ellagitannins	ellagic acid	301	301, 447, 463, 477	2.1
	ellagic acid *O*-pentoside	433		
	ellagic acid *O*-hexoside	463		
	methylellagic acid *O*-hexoside	477		
	ellagic acid *O*-(*O*-galloyl)-hexoside	615		
	methyellagic acid *O*-pentoside isomer I–II	447		
	methylellagic acid *O*-glucuronide	491		
	agrimoniin	934		
Flavonoids, chalcones and related polyphenols	phlorizin	435	273, 287, 449	1.0
	unknown flavonoid derivative	575		
Phenolic acids and derivatives	protocatechuic acid *O*-hexoside	315	153, 179	n/a
	gallic acid derivative	377		
Triterpenoids and derivatives	tormentic acid isomer I–II	487	469, 487	28.9
	tormentic acid *O*-hexoside isomer I–II	695		
	trihydroxy-urs-12-en-28-oic acid-28-*O*-hexoside isomer	695		
	trihydroxy-urs-12-en-28-oic acid isomer	487		
	dihydroxy-oxo-urs-12-en-28 oic acid-*O*-hexoside isomer I–II	693		
	dihydroxy-oxo-urs-12-en-28 oic acid isomer	485		
	tetrahydroxy-methoxy-urs-12-en-28-oic acid-28-*O*-hexoside	695		
	myrianthic acid	503		
	cecropiacic acid	517		
	undefined triterpenoic acid hexoside	679		
Unknown compounds	unknown	441	n/a	n/a
	unknown	511		
	unknown	711		
	unknown	605		
	unknown	507		
	unknown	493		
	unknown	493		
	unknown	507		

## Data Availability

The raw data supporting the conclusions of this article will be made available by the authors on request.

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
