# Peer review of "In Vitro Evaluation of Antioxidant and Cytokine-Modulating Activity of Tormentil Rhizome Extract and Its Microbial Metabolites in Human Immune Cells"

_ijms, 2025, doi:10.3390/ijms262211164_

Round 1
Reviewer 1 Report
Comments and Suggestions for Authors
The article is well-written. I have a few minor suggestions/comments:
Numerous editing errors should be corrected: missing italics, spaces, punctuation, etc.
Line 138 and others related to this issue: procyanidins are derivatives of flavan-3-ol.
Lines 145 and 146: Phlorizin belongs to the chalcone subgroup. Writing that it is a flavonoid would also suggest that flavan-3-ols be classified as flavonoids.
"Protocatechuic acid derivatives" or "protocatechuic acid derivative"?
On what basis did the authors choose the concentration range for biological activity testing?
How many donors were recruited for the human neutrophil model studies?
Author Response
Response to Reviewer
1. Numerous editing errors should be corrected: missing italics, spaces, punctuation, etc.
I thank the reviewer for pointing this out. I have carefully revised the entire manuscript and corrected the identified formatting and editing issues, including missing italics, spaces, punctuation, and other typographical errors.
2. Line 138 and others related to this issue: procyanidins are derivatives of flavan-3-ol.
I thank the reviewer for this accurate comment. I have corrected the wording to reflect that procyanidins are oligomers of flavan-3-ols (condensed tannins), rather than derivatives. The revised text now uses the appropriate terminology.
3. Lines 145 and 146: Phlorizin belongs to the chalcone subgroup. Writing that it is a flavonoid would also suggest that flavan-3-ols be classified as flavonoids.
I thank the reviewer for this precise observation. I agree that phlorizin is a dihydrochalcone and should not be classified as a flavonoid. I have corrected the terminology in the text accordingly and updated Table 1 by renaming the compound class to “Flavonoids, chalcones and related polyphenols”, which accurately reflects the chemical nature of the listed constituents.
4. “Protocatechuic acid derivatives” or “protocatechuic acid derivative”?
I thank the reviewer for this clarification. I have adjusted the terminology accordingly: when referring to a single compound, the text now uses “protocatechuic acid derivative”, and when describing the entire group, “protocatechuic acid derivatives”. The wording has been corrected in the revised manuscript.
5. On what basis did the authors choose the concentration range for biological activity testing?
I thank the reviewer for this important question. The concentration range used in the biological assays was selected based on my previous experience with structurally similar, polyphenol-rich plant extracts. In earlier studies, higher concentrations of such extracts frequently led to non-specific cytotoxic effects, particularly in neutrophils and THP-1 macrophages, which limited reliable interpretation of functional endpoints. Therefore, I chose a concentration range that had been shown in comparable experiments to be well tolerated while still allowing detection of biologically meaningful changes in ROS production and cytokine secretion.
This clarification has been added to the Materials and Methods section.
In Section 4.9 (neutrophils), I included the following sentence:
“The concentration range was selected based on previous studies using structurally similar polyphenol-rich plant extracts, in which higher doses frequently caused non-specific cytotoxic effects in immune cell models [30-37].”
In Section 4.13 (THP-1 macrophages), I added:
“The same concentration range was used as in neutrophil assays, as it had previously been shown to be non-cytotoxic for polyphenol-rich plant extracts [30,37].”
6. How many donors were recruited for the human neutrophil model studies?
I thank the reviewer for this question. Three healthy adult donors were recruited for the human neutrophil model studies. This information has now been added to Section 4.8 of the revised manuscript.
Reviewer 2 Report
Comments and Suggestions for Authors
Overall, the manuscript has no major issues and is constructed adequately with reasonable assays. A few comments and suggestions:
- Spelling mistake in line 135 "presented studies"; line 218 "containing", and several other areas. Please check through the manuscript and make corrections accordingly.
- Figure 3 please remake the figure to show the raw flow cytometry data to supplement the bar chart.
Author Response
Response to Reviewer
1. “Spelling mistake in line 135 ‘presented studies’; line 218 ‘containing’, and several other areas. Please check through the manuscript and make corrections accordingly.”
I thank the reviewer for pointing out these language issues. I have corrected the spelling mistakes in lines 135 and 218 and carefully rechecked the entire manuscript to identify and fix additional typographical and grammatical errors. The revised version now reflects consistent and accurate wording throughout.
2. “Figure 3 please remake the figure to show the raw flow cytometry data to supplement the bar chart.”
I thank the Reviewer for this valuable comment. In response, I have improved the clarity of Figure 3 by adding representative raw flow cytometry plots as a separate panel.
Reviewer 3 Report
Comments and Suggestions for Authors
- The abstract states that "IL-8 showed moderate induction," but the results (Fig 5) show a very pronounced induction (120-200% of control), which is a major finding. The wording should be adjusted to more accurately reflect the strength of this effect (e.g., "marked induction" or "was significantly enhanced").
- Figure 3 is difficult to read. Consider showing only the highest concentration to make it easier to read.
- In section 2.3.3, you wrote that a fraction reduced "IL-1β," but the graph shows it was "IL-8." Please correct this.
- Double-check that the text correctly states that you used LPS for cytokine experiments and f-MLP for the ROS experiments.
- When talking about the increase in IL-8, focus more on what it means for immune cells (like attracting more neutrophils) rather than just referring to skin cell studies.
Author Response
Response to Reviewer
1. “The abstract states that ‘IL-8 showed moderate induction,’ but the results (Fig 5) show a very pronounced induction (120–200% of control), which is a major finding. The wording should be adjusted to more accurately reflect the strength of this effect.”
I thank the reviewer for this important observation. I agree that the wording in the abstract understated the magnitude of the IL-8 response. As suggested, I have revised the text to accurately reflect the pronounced induction observed in the results. The phrase “moderate induction” has been replaced with wording that clearly indicates a marked and significant increase.
Revised fragment:
“EtTR and all TRGM fractions significantly reduced ROS production, while the extract and selected metabolites decreased IL-1β and TNF-α secretion in neutrophils, whereas IL-8 showed marked induction.”
2. “Figure 3 is difficult to read. Consider showing only the highest concentration to make it easier to read.”
I thank the Reviewer for this valuable comment and acknowledge that showing only the highest concentration would indeed improve visual clarity. However, I decided to retain all tested concentrations in Figure 3 because omitting the intermediate doses would make this experiment inconsistent with the other assays in the manuscript, which are all presented across the same full concentration range. Moreover, another Reviewer explicitly requested the inclusion of additional cytometry data, and therefore reducing the figure further would conflict with that requirement. To balance clarity with methodological coherence and to meet both Reviewers’ recommendations, I kept the full dataset in the main figure while providing representative raw cytometry plots in an additional panel.
To address the reviewer’s concern, I have included the raw flow cytometry plots corresponding to these data in the Supplementary Materials to provide full transparency and facilitate interpretation.
3. “In section 2.3.3, you wrote that a fraction reduced ‘IL-1β,’ but the graph shows it was ‘IL-8.’ Please correct this.”
I thank the reviewer for noticing this error. In section 2.3.3, the cytokine name was indeed incorrectly stated as “IL-1β,” while the corresponding graph shows results for “IL-8.” I have corrected this in the revised manuscript to ensure consistency and accuracy.
4. “Double-check that the text correctly states that you used LPS for cytokine experiments and f-MLP for the ROS experiments.”
I thank the reviewer for this helpful comment. I have double-checked the manuscript and confirmed that the text correctly states the use of LPS for cytokine measurements and f-MLP for the ROS experiments. The relevant sections have now been revised to ensure full clarity and consistency.
5. “When talking about the increase in IL-8, focus more on what it means for immune cells (like attracting more neutrophils) rather than just referring to skin cell studies.”
I thank the reviewer for this insightful comment. I have revised the discussion to emphasize the immunological relevance of IL-8 induction, specifically its role in neutrophil recruitment and activation, rather than focusing solely on keratinocyte studies. The revised text highlights the consequences of IL-8 upregulation for innate immune cells.
Revised fragment:
“In innate immune cells, increased IL-8 secretion has a direct functional consequence, as IL-8 is one of the major chemokines responsible for neutrophil recruitment and activation. Even moderate elevations of IL-8 can enhance neutrophil chemotaxis, promote firm adhesion to the endothelium, and prime these cells for oxidative burst and degranulation. Therefore, the IL-8 increase observed in our neutrophil model may reflect an adaptive chemotactic response aimed at improving early immune surveillance rather than a purely pro-inflammatory effect. Such controlled IL-8 upregulation is consistent with a transient, redox-linked signaling shift that prepares neutrophils for environmental stress without inducing excessive inflammatory damage [33,34].”